# Reproducibility of individual effect sizes in meta-analyses in psychology

**Esther Maassen**[1]*, **Marcel A. L. M. van Assen**[1,2], **Michèle B. Nuijten**[1], **Anton Olsson-Collentine**[1], **Jelte M. Wicherts**[1]

1 Department of Methodology and Statistics, Tilburg University, Tilburg, the Netherlands, 2 Department of Sociology, Utrecht University, Utrecht, the Netherlands

* emaassen@protonmail.com

**Data Availability Statement:** All data files are available from the Open Science Framework database (https://osf.io/7nsmd/).

## Abstract

To determine the reproducibility of psychological meta-analyses, we investigated whether we could reproduce 500 primary study effect sizes drawn from 33 published meta-analyses based on the information given in the meta-analyses, and whether recomputations of primary study effect sizes altered the overall results of the meta-analysis. Results showed that almost half ($k = 224$) of all sampled primary effect sizes could not be reproduced based on the reported information in the meta-analysis, mostly because of incomplete or missing information on how effect sizes from primary studies were selected and computed. Overall, this led to small discrepancies in the computation of mean effect sizes, confidence intervals and heterogeneity estimates in 13 out of 33 meta-analyses. We provide recommendations to improve transparency in the reporting of the entire meta-analytic process, including the use of preregistration, data and workflow sharing, and explicit coding practices.

## Introduction

The ever-increasing growth of scientific publication output [1] has increased the need for -and use of- systematic reviewing of evidence. Meta-analysis is a widely used method to synthesize quantitative evidence from multiple primary studies. Meta-analysis involves a set of procedural and statistical techniques to arrive at an overall effect size estimate, and can be used to inspect whether study outcomes differ systematically based on particular study characteristics [2]. Careful considerations are needed when conducting a meta-analysis because of the many (sometimes arbitrary) decisions and judgments that one has to make during various stages of the research process [3]. Procedural differences in coding primary studies could lead to variation in results, thus potentially affecting the validity of drawn conclusions [4]. Likewise, meta-analysts often need to perform complex computations to synthesize primary study results, which increases the risk of faulty data handling and erroneous estimates [5]. When these decisions and calculations are not carefully undertaken and specifically reported, the methodological quality of the meta-analysis cannot be assessed [6,7]. Additionally, reproducibility (i.e., reanalyzing the data by following reported procedures and arriving at the same result) is undermined by reporting errors and by inaccurate, inconsistent, or biased decisions in calculating effect sizes.

**Funding:** This research was supported by a Consolidator Grant 726361 (IMPROVE) from the European Research Council (ERC, https://erc. europa.eu), awarded to J.M. Wicherts. The funders had no role in study design, data collection and analysis, decision to publish, or preparation of the manuscript.

**Competing interests:** I have read the journal's policy and the authors of this manuscript have the following competing interests: Jelte M. Wicherts is a PLOS ONE Editorial Board member. This does not alter the authors' adherence to PLOS ONE Editorial policies and criteria.

Research from various fields has demonstrated issues arising from substandard reporting practices. Transformations from primary study effect sizes to the standardized mean difference (SMD) used in 27 biomedical meta-analyses were found to be commonly inaccurate [8], leading to irreproducible pooled meta-analytic effect sizes. Randomized controlled trials often contain an abundance of outcomes, leaving many opportunities for cherry-picking outcomes that can influence conclusions in various ways [9,10]. Similar evidence of incomplete and nontransparent meta-analytic reporting practices was found in the organizational sciences [11–13]. However, the numerous choices and judgment calls meta-analysts need to make do not always influence meta-analytic effect size estimates [14]. Finally, a severe scarcity of relevant information related to effect size extraction, coding, and adhering to reporting guidelines hinders and sometimes obstructs reproducibility in psychological meta-analyses [15].

This project builds upon previous efforts examining reproducibility of psychological primary study effect sizes and associated meta-analytic outcomes. In the first part, we considered effect sizes reported in 33 randomly chosen meta-analytic articles from psychology, and searched for the corresponding primary study articles to examine whether we could recompute 500 effect sizes reported in the meta-analytic articles; we refer to this as *primary study effect size reproducibility*. In the second part, we considered if correcting any errors in these primary study effect sizes affected main meta-analytic outcomes. Specifically, we looked at the estimate of the average effect size and its confidence interval, and heterogeneity parameter $\tau^2$. We refer to this as *meta-analysis reproducibility*. Although we acknowledge that more aspects in a meta-analysis could be checked for reproducibility (e.g., search strategy, application of inclusion and exclusion criteria), we focus here on the effect sizes and analyses thereof. While primary study effect size reproducibility is important for assessing the accuracy of the specific effect size calculations, meta-analytic reproducibility bears on the overall conclusions drawn from the meta-analysis. Without appropriate reporting on what was done and what results were found it is untenable to determine the validity of the meta-analytic results and conclusion [16].

## Part 1: Primary study effect size reproducibility

In Part 1, we documented primary study effect sizes as they are reported in meta-analyses (i.e., in a data table) and attempted to reproduce them based on the calculation methods specified in the meta-analysis and the estimates reported in the primary study articles. There exist several reasons why primary study effect sizes might not be reproducible. First, the primary study article may lack sufficient information to reproduce the effect size (e.g., missing standard deviations). Second, it might be unclear which information from the primary study was used to compute the effect size. That is, multiple statistical results may be reported in the paper, and ambiguous reporting in the meta-analytic paper might obscure which information from the primary study was used in the computation or which calculation steps were performed to standardize the effect size. Finally, it could also be that retrieval, calculation, or reporting errors were made during the meta-analysis.

We hypothesized that a sizeable proportion of reproduced primary effect sizes would be different from the original calculation of the authors because of ambiguous reporting or errors in effect size transformations [8]. We expected more discrepancies in effect size estimates that require more calculation steps (i.e., SMDs) compared to effect sizes that are often extracted directly (i.e., correlations). We also expected more errors in unpublished primary studies compared to published primary studies, because the former are less likely to adhere to strict reporting standards and are sometimes not peer-reviewed. Our goal was to document the percentage of irreproducible effect sizes and to categorize these irreproducible effect sizes as being

*incomplete* (i.e., not enough information is available), *incorrect* (i.e., an error was made), or *ambiguous* (i.e., it is unclear which effect size or standardization was chosen).

The hypotheses, design, and analysis plan of our study were preregistered and can be found at https://osf.io/v2m9j. In this paper, we focus only on primary study effect size and meta-analysis reproducibility. Additional preregistered results concerning reporting standards can be found in S1 File (https://osf.io/pf4x9/). We deviated from our preregistration in some ways. First, we extended our preregistration by also checking whether irreproducible primary study effect sizes affected the meta-analytic estimate of heterogeneity $\tau^2$, in addition to the already preregistered meta-analytic pooled effect size estimate and its confidence interval. We also checked the differences in reproducibility of primary study effect sizes that we classified as outliers compared to those we classified as non-outliers, which was not included in the preregistration. Another deviation from the preregistration was that we did not explore the between-study variance across meta-analyses with and without moderators. The reason for this is that the $Q$ and $I^2$ statistics tend to be biased in certain conditions when true effect sizes are small and when publication bias and the number of studies is large [17], and we did not take that into account when we preregistered our study.

## Method

### Sample

**Meta-analysis selection.**   The goal of the meta-analysis selection was to obtain a representative sample of psychological meta-analyses. We therefore included a sample from the PsycArticles database from 2011 and 2012 that matched the search criteria "*meta-a\**", "*research synthesis*", "*systematic review*", or "*meta-anal\**". Only meta-analyses that contained a data table with primary studies, sample sizes and effect sizes, and had at least ten primary studies were included. Earlier research by Wijsen (https://osf.io/xswvg/) and [18] already drew random samples from the eligible meta-analytic articles, resulting in 33 meta-analyses. For this study we used the same sample of 33 meta-analyses to assess reproducibility. A list of meta-analyses, a detailed sampling scheme and flowcharts can be found in S2 File: https://osf.io/43ju5/.

In total, we selected 33 meta-analytic articles containing 1,978 primary study effect sizes, of which we sampled 500 (25%) primary study effect sizes to reproduce. We decided on 500 primary study effect sizes because of feasibility constraints given the substantial time needed to fully reproduce effect sizes. The coding process entailed selecting, retrieving, and recomputing each effect size by two independent coders (EM and AOC). Differences or disagreements in coding were mostly due to both coders selecting different effects from the primary study, due to ambiguous reporting in the meta-analysis. These disagreements were solved after discussion and one primary study effect size was chosen. The interrater reliability for which category the effect size belonged to (i.e., a reproduced, different, incomplete or ambiguous effect) was $\kappa$ = .71 [19]. We estimate the total time spent on selecting, retrieving, recomputing, verifying, and discussing all primary study effect sizes to be over 500 hours.

**Primary study selection.**   We wanted to ensure that our sample of primary studies would give the most accurate reflection of the reproducibility of primary effect sizes. There are specific primary study characteristics that may relate to reproducibility. For instance, some data errors in meta-analyses are likely to result in outliers that inflate variance estimates [20]. Since outliers occur less often than non-outliers and we wanted to ensure a fair distribution of types of effect sizes, we oversampled primary study effect sizes that could be considered outliers compared to the rest of the primary studies in the meta-analysis. We later corrected for this oversampling. More information on how we classified outlier primary study effect size is given

in S2 File: https://osf.io/43ju5/. Detailed information on how we corrected for oversampling, including an example, is displayed in S5 File: https://osf.io/u2j3z/.

For each meta-analysis separately, we first fitted a random-effects model in which we used the $Q$ statistic from the leave-one-out-function in the *metafor* package (version 1.9–9) in $R$ (version 3.3.2) to classify all primary studies with their reported effect sizes as either being an outlier or non-outlier [21,22]. If the $Q$ statistic after the leave-one-out function showed a statistically significant ($\alpha$ = 0.05 for all analyses) difference from the $Q$ statistic from the complete meta-analysis, the left out primary study effect size was classified as an outlier. We then randomly sampled ten outlier primary study effect sizes per meta-analysis, if that many could be obtained. The median number of outlier primary effect sizes was seven. In total, we selected 197 outlier primary study effect sizes from 33 meta-analyses, leaving 303 effect sizes that remained to be sampled from the non-outlier primary study effect sizes. We randomly selected approximately ten non-outlier primary study effect sizes per meta-analysis. If fewer than ten could be selected, we randomly divided the remaining number among other meta-analyses that had effect sizes left to be sampled.

## Procedure

To recompute the primary study effect sizes, we strictly followed calculation methods as described in the meta-analytic article. In cases where the meta-analysis was unclear as to which methods were used, we employed well-known formulas for effect size transformation. The information we extracted and formulas we used to compute and transform all primary study effect sizes are displayed for each meta-analysis in S3 File: https://osf.io/pqt9n/. In case of a two-group design we assumed equal group sizes if no further information was given in the meta-analysis or the primary study article. We adjusted all effects in such a way that insofar the prediction of the meta-analysts was corroborated, the mean effect size would be positive, which entailed changing the sign of the effect sizes of five meta-analyses. Consequently, a negative effect size indicates an effect opposite to what was expected. We analyzed results using the *metafor* package (version 2.1–0) in $R$ (version 3.6.0) [21,22]. Relevant code and files to reproduce the results from this study can be found at https://osf.io/7nsmd/files/.

Our procedure for recalculating primary study effect sizes is displayed in Fig 1. We categorized reproduced primary study effects into one of four categories we considered, ranging from best to worst outcome: (0) *reproducible*: we could reproduce the effect size as reported in the meta-analytic article or with a margin of error of correlation $r <$ .025 or Hedges' $g <$ .049; (1) *incomplete*: not enough information was available to reproduce the effect size (e.g., SDs are missing in the primary article). In this case we copied the original effect size as reported in the meta-analysis because we were not sure what computations the meta-analysts performed, or whether they had contacted the authors for necessary statistics; (2) *incorrect*: our recalculation resulted in a different effect size of at least $r \geq$ .025 or Hedges' $g \geq$ .049 (i.e., a potential calculation or reporting error was made); or (3) *ambiguous*: it was unclear what steps the meta-analysts followed so we categorize the effect as ambiguous. We consider ambiguous effects (category 3) to be more problematic than incorrect effects (category 2), as for the incorrect effects it was clear which effect size the meta-analysts had chosen, whereas for the ambiguous effects it was impossible to ascertain if we had selected the correct effect size, and what steps were taken to get to the effect size as reported. We acknowledge this categorization is open to discussion, and only used it descriptively throughout this study. When we found multiple appropriate effects given the information available in the meta-analysis, we chose the primary study effect size that showed the least discrepancy with the effect size as reported in the meta-analysis.

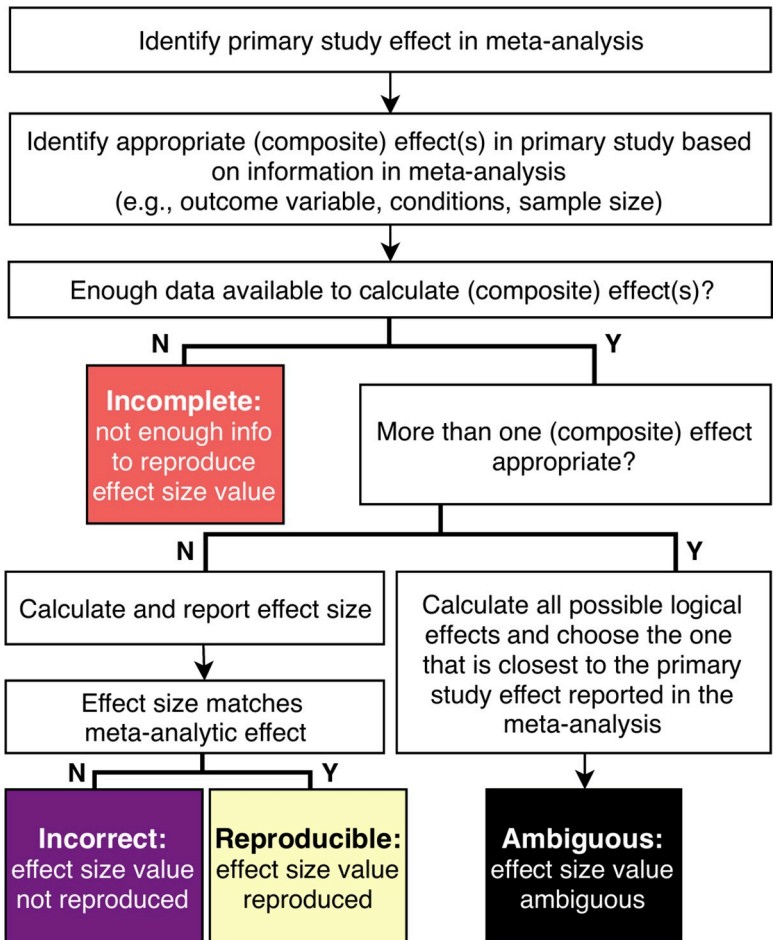

**Fig 1. Decision tree of primary study effect size recalculation and classification of discrepancy categories.** A composite effect refers to a combination of two or more effects.

Next to checking whether certain primary study effect sizes were irreproducible due to incomplete, erroneous, or ambiguous reporting, we were also interested in quantifying whether the discrepancy between the reported and reproduced effect size estimates were small, moderate, or large. Effect sizes in various psychological fields (e.g., personality, social, developmental, clinical psychology, intelligence) show small, moderate, and large effect sizes corresponding approximately to $r = 0.10, 0.25$, and $0.35$ [23,24]. Based on these results, we chose the classifications of discrepancies in correlation $r$ to be small [$\geq 0.025, <0.075$], moderate [$\geq 0.075, <0.125$] and large [$\geq 0.125$] and transformed them to similar classifications for the other effect sizes based on $N = 64$, relating to the 50th percentile of the degrees of freedom of reported test statistics in eight major psychology journals [24]. For Hedges' $g$, classifications were small [$\geq .049, < .151$], moderate [$\geq .151, < .251$], and large [$\geq .250$]. For Cohen's $d$, classifications were small [$\geq .050, < .152$], moderate [$\geq .152, < .254$] and large [$\geq .254$]. For Fisher's $r$-to-$z$ transformed correlation, classifications were the same as $r$.

## Results

Out of 500 sampled primary study effect sizes we could reproduce 276 without any issues (*reproducible*: 55%). For 54 effect sizes, the primary study paper did not contain enough

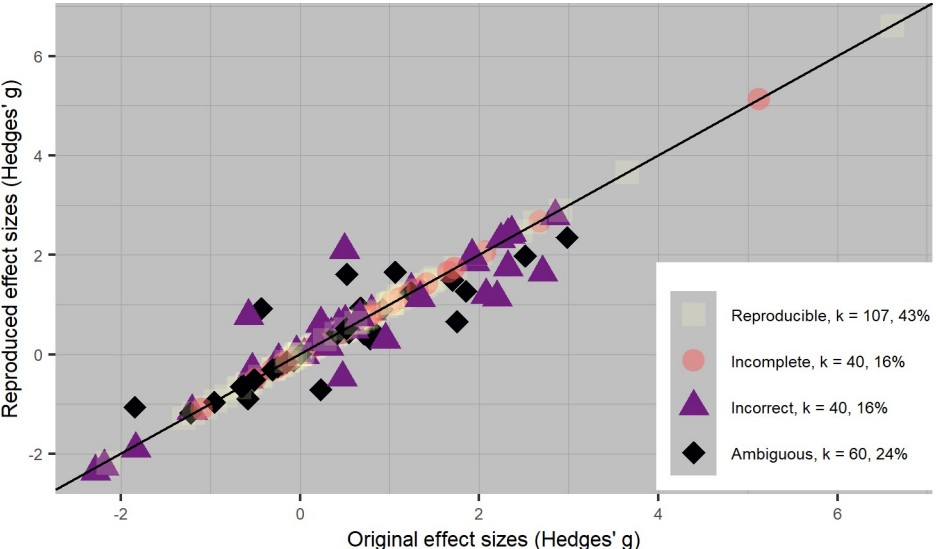

**Fig 2. Scatterplot of 247 original and reproduced standardized mean difference effect sizes from 33 meta-analyses.** All effect sizes are transformed to Hedges' *g*.

information to reproduce the effect (*incomplete*: 11%), so the original effect size was copied. For 74 effect sizes, a different effect than originally stated in the meta-analytic article was calculated (*incorrect*: 15%), whereas for 96 effect sizes it was unclear what procedure was followed by the meta-analysts (*ambiguous*: 19%), and the effect size which was most relevant and closest to the reported effect size was chosen.

Fig 2 displays all primary study effect sizes that used either Cohen's *d* or Hedges' *g* as the meta-analytic effect size (*k* = 247), where all primary study effect sizes in Cohen's *d* were transformed to Hedges' *g*. The horizontal axis displays the original reported effect sizes and the vertical axis the reproduced effects; all data points on the diagonal line indicate no discrepancy between reported and reproduced primary study effect sizes. Likewise, Fig 3 displays all

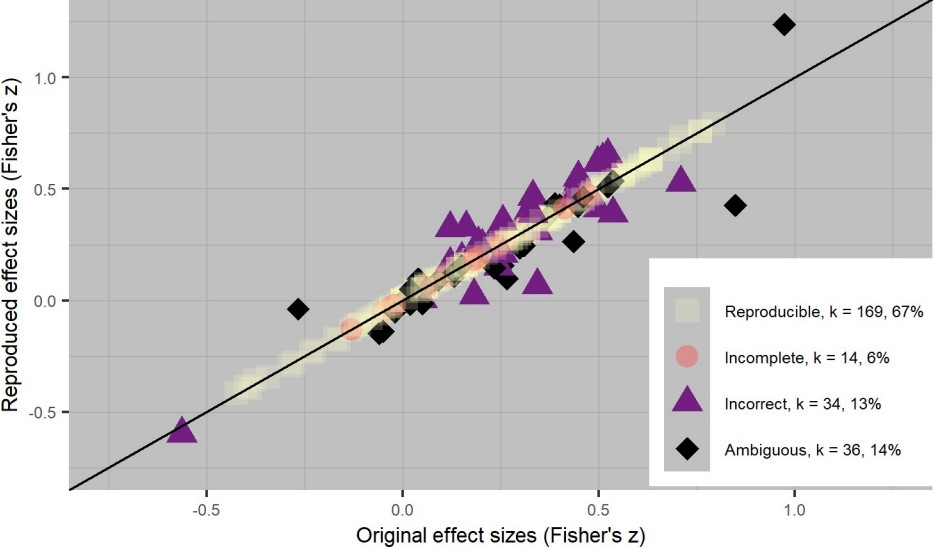

**Fig 3. Scatterplot of 253 original and reproduced correlation effect sizes from 33 meta-analyses.** All effect sizes are transformed to Fisher's *z*.

primary study effect sizes that used either a product-moment correlation $r$ or Fisher's $r$-$to$-$z$ transformed correlation ($k = 253$), where all primary study effect sizes using a product-moment correlation $r$ were transformed to Fisher's $z$ correlation. The results as illustrated in Figs 2 and 3 can be found separately per meta-analysis in S4 File: https://osf.io/65b8z/.

In total, 114 out of 500 recalculated primary study effect sizes (23%) showed effect size discrepancies compared to the primary study effect sizes as reported in the meta-analytic articles. Of those 114 discrepancies, 62 were small (54%), 21 were moderate (18%), and 31 were large (27%). We note that it is possible for a primary study effect size to be classified as irreproducible, even if we found no discrepancies between the reported and recalculated primary study effect size estimate. This happened for instance for all primary studies that did not contain enough information to reproduce the effect size, for which we copied the reported effect size. The number of effect sizes we calculated that were larger than reported ($k = 162$) was approximately equal to those that we calculated that were smaller than reported ($k = 165$), which indicates no systematic bias in either direction.

The most common reason for not being able to reproduce a primary study effect size was missing or unclear information in the meta-analysis (i.e., ambiguous effect sizes, $k = 96$, 19%). More specifically, it was often unclear which specific effect was extracted from the primary study because multiple effects were relevant to the research question, and we did not know if and how the included effect was constructed from a combination of multiple effects. Other prevalent issues pertaining to potential data errors or lack of clarity in the meta-analytic process were inconsistencies and unclear reporting in inclusion criteria within a meta-analytic article ($k = 67$), uncertainty about which samples or time points were included ($k = 50$), reporting formulas or corrections that were incorrect or not used ($k = 23$), including the same sample of respondents for multiple effects without correction ($k = 7$), lacking information on how the primary study effect was transformed to the effect size included in the meta-analysis ($k = 3$), and mistaking the standard error for the standard deviation when calculating effects ($k = 2$).

Within over a quarter of primary studies (147; 29%, see Table 1) we combined multiple effects into one overall effect size estimate for that primary study. The percentage of irreproducible effect sizes is relatively large within this group. Within this subset, 18% was classified as incorrect (single effect sizes: 13%), 34% as ambiguous (single effect sizes: 13%), 1% as incomplete (single effect sizes: 15%), and 46% as reproducible (single effect sizes: 59%); $X^2$ (3, $N = 500$) = 45.78, $p < .0001$, $\Phi_{Cramer} = 0.30$, showing that combining multiple effect sizes into one overall estimate is moderately associated with irreproducibility of effect size estimates.

Table 2 contains descriptive statistics on reproducibility and various primary study characteristics. We hypothesized that primary studies with SMDs would be less reproducible than studies with other effect sizes. In line with our hypothesis, we found that 57% of all primary studies containing SMDs were irreproducible, whereas for primary studies with correlations this was 33% (see Table 2). This difference between SMDs and correlations can also clearly be seen in Fig 2 and Fig 3. Constructing SMDs requires more transformations and calculations of

**Table 1. Reproducibility frequencies separated by primary study effect sizes consisting of one or multiple combined effect sizes.**

|  | Single effect size | Combined effect sizes |
|---|---|---|
| Reproducible | 208 (59%) | 68 (46%) |
| Incorrect | 47 (13%) | 27 (18%) |
| Incomplete | 52 (15%) | 2 (1%) |
| Ambiguous | 46 (13%) | 50 (34%) |
| Total | 353 (100%) | 147 (100%) |

**Table 2. Reproducibility frequencies separated by primary study characteristics.**

|  | Irreproducible | Reproducible | Total |
|---|---|---|---|
| SMD | 140 (57%) | 107 (43%) | 247 (100%) |
| Correlation | 84 (33%) | 169 (67%) | 253 (100%) |
| Outlier | 77 (39%) | 120 (61%) | 197 (100%) |
| Non-outlier | 147 (49%) | 156 (51%) | 303 (100%) |
| Published | 216 (47%) | 239 (53%) | 455 (100%) |
| Unpublished | 8 (18%) | 37 (82%) | 45 (100%) |

effects, whereas correlations are often extracted from the primary paper as is. As such, it is not surprising that we found that correlations are more reproducible than SMDs in our sample.

We hypothesized that effect sizes from unpublished studies would be less reproducible compared to published studies, but contrary to our expectation we found that 18% of unpublished studies and 47% of published studies were irreproducible (see Table 2).

Because we oversampled primary study effect sizes classified as outliers, our sample of 500 primary study effect sizes is not representative for the 1,978 effect sizes we sampled from. In the sample of 1,978 effect sizes, we classified 30% as an outlier, compared to 39% in our sample of 500. This means our sample contains too many outlier primary study effect sizes, and too few non-outliers. To calculate the probability that any given effect size in a certain meta-analysis is irreproducible, we needed to correct for this overrepresentation of outliers by design (S5 File: https://osf.io/u2j3z/). We calculated correction weights using type of effect size (outlier or non-outlier) as the auxiliary variable, and used the sample proportions of outliers and non-outliers to determine the probability of finding a potential error (i.e., either an incomplete, ambiguous, or different primary study effect size) for each meta-analysis and across all 33 meta-analyses in total. The computed primary study error probability for the 33 meta-analyses varied from 0 to 1. Across all meta-analyses we estimated the chance of any randomly chosen primary study effect size being irreproducible to be 37%.

Fig 4 displays a bar plot with the frequency of irreproducible effect sizes per meta-analysis. The distribution of reproducible effect sizes (category 0) ranged from 0% to 100% with a mean

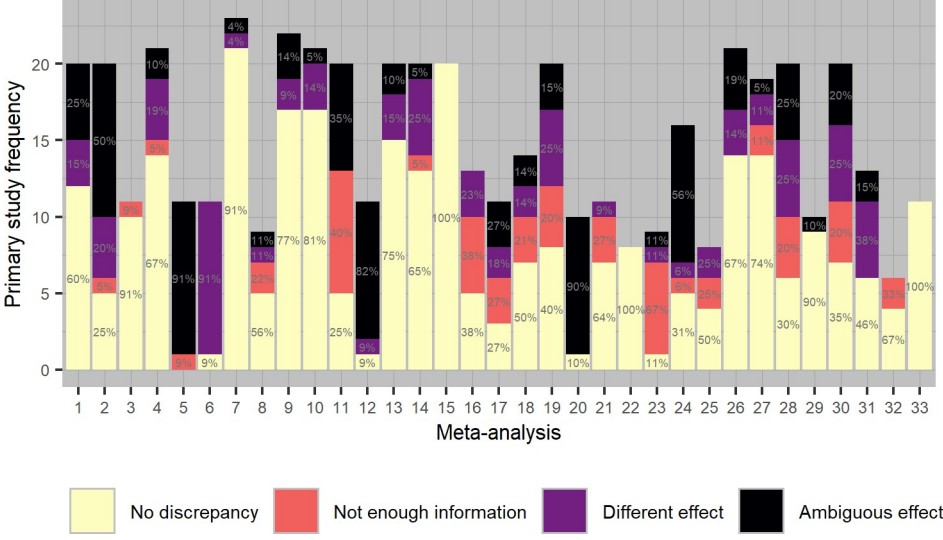

**Fig 4. Frequencies of reproduced primary study effect sizes with and without errors, per meta-analysis.**

of 53% and median of 56%. Only three of the 33 samples of primary studies were completely reproducible, and one was completely irreproducible ($k = 11$). The percentage of incorrect effect sizes (category 1) ranged from 0% to 91% across meta-analyses, (mean = 14%, median = 11%); incomplete effect sizes ranged from 0% to 67% (mean = 12%, median = 5%), and ambiguous effect sizes (category 3) ranged from 0% to 91% (mean = 19%, median = 11%). Note that the reporting within meta-analyses is often at least partly ambiguous (24 out of 33 meta-analyses contain at least one ambiguous effect size).

### Exploratory findings

The previously reported results can be considered confirmatory because we preregistered our hypotheses and procedures. Next to confirmatory analyses we also performed one exploratory analysis, in which we compared the reproducibility of outlier and non-outlier primary study effect sizes. We found that 39% of all outlier effect sizes were irreproducible, whereas for non-outlier effect sizes it was 49% (see also Table 2).

## Conclusion

In Part 1 we set out to investigate the reproducibility of 500 primary study effect sizes as reported in 33 psychological meta-analyses. Of the 500 reported primary study effect sizes, almost half (224) could not be reproduced, and 30 out of 33 meta-analyses contained effect sizes that could not be reproduced. Poor reproducibility at the primary study level might affect meta-analytic outcomes, which is worrisome because it could bias the meta-analytic evidence and lead to substantial changes in conclusions. We investigate this in Part 2 of this study.

## Part 2. Meta-analysis reproducibility

In Part 2, we examined whether irreproducible primary study effect sizes affect three meta-analytic outcomes; the overall effect size estimates, its confidence interval, and the estimate of heterogeneity parameter $\tau^2$. We hypothesized to find discrepancies between the reported and reproduced primary studies, and subsequently also expected several meta-analytic pooled effect size estimates to be irreproducible. Discrepancies in primary study effect sizes we found in Part 1 can be either systematic or random. Systematic errors in primary study effect sizes will bias results and thus have a larger impact on the meta-analytic mean estimate, whereas random primary study errors can be expected to increase the estimate of heterogeneity in the meta-analysis. Because we expected most primary study errors to be random instead of systematic, we hypothesized corrected primary study effect sizes would have a larger impact on the boundaries of the confidence interval (i.e., smaller CIs after adjustment) than the meta-analytic effect size estimate.

## Method

To investigate meta-analysis reproducibility, we first documented for each of the 33 meta-analyses which software was used to run the meta-analysis, what type of meta-analysis was conducted (i.e., fixed effect, random-effects, or mixed-effects), and what kind of estimator was originally used. The information we extracted to reproduce the meta-analytic effects can be found for each meta-analysis in S3 File: https://osf.io/pqt9n/. For each of the 33 meta-analyses, we conducted two analyses: one with the primary study effect sizes as reported in the meta-analysis, and one with the primary study effect sizes as we recalculated them. We then documented the pooled meta-analytic effect size estimate, its confidence interval, and the estimated

$\tau^2$ parameter for both the reported and reproduced meta-analysis, and compared these outcomes for discrepancies. For Part 2 we upheld the same discrepancy measures as in Part 1.

The results presented next are based on the subset of primary study effect sizes that were sampled for checking, instead of complete meta-analyses including all primary studies. We decided to report on subsets of the meta-analysis containing only sampled studies, because the effect of corrected primary study effect sizes on meta-analytic outcomes can best be shown when only the corrected primary study effect sizes are included in the meta-analysis. We also conducted analyses on all 33 complete meta-analyses, for which results are reported in S1 File: https://osf.io/pf4x9/.

## Results

We first documented which procedures the meta-analysts used for their analyses. We found the same level of imprecise reporting here as in Part 1: most meta-analytic articles reported scarce information related to estimation methods. Many meta-analyses simply referred to well-known meta-analysis books without mentioning specifically which method was used. Out of 33 meta-analyses, only two explicitly reported which models, software and estimator they used for their estimation. For the other meta-analyses, we were forced to guess the used estimation method. Most meta-analytic authors ($m = 25$, 76%) used either a random-effects or both fixed effect and random-effects models.

For meta-analytic outcomes, 13 out of 33 meta-analyses (39%) showed discrepancies in either the pooled effect size estimate, its confidence interval, or $\tau^2$ parameter. For example, in meta-analysis no. 17 the pooled effect size estimate the subset we sampled was $g = 0.35$, 95% CI [-0.02, 0.72], which dropped to $g = 0.23$, 95% CI [-0.01, 0.47] after three (out of 11) primary study effect sizes showed large discrepancies between the reported and recalculated results. Note that even though the number of primary study effect sizes that were irreproducible was large, the number of discrepancies between the reported and reproduced primary study effect sizes was relatively small, mostly due to our decisions to copy the effect size if not enough information was available, or choosing the estimate that most closely resembled the reported one when the effect was ambiguous. We found small discrepancies in the pooled effect size estimates for nine out of 33 meta-analyses (27%), displayed in panel *a* of Fig 5 (for all meta-analyses using SMDs), and Fig 6 (for all meta-analyses using correlations). We plotted the difference between the upper and lower bound of the confidence intervals in Figs 5 and 6 (panel *b*). We found 13 meta-analyses with discrepancies in the confidence intervals (39%), of which nine were small (Hedges' $g$: $\geq .049$ and $< .151$, Fisher's $z$: $\geq .025$ and $< .075$) and three were moderate (Hedges' $g$: $\geq .151$ and $< .251$, Fisher's $z$: $\geq .075$ and $< .125$). In line with our hypothesis, this result shows that corrected primary study effect sizes have a larger impact on the boundaries of the confidence interval than the pooled effect size estimate. In none of the meta-analyses was the statistical significance of the average effect size affected by using the recalculated primary study effect sizes.

We did not find any evidence for systematic bias in meta-analytic results; we estimated 19 pooled effect sizes to be larger than originally reported and 14 to be smaller than originally reported. We estimated wider pooled effect size CIs in 18 cases and smaller CIs in 15. This latter result is contrary our expectations that CI estimates would be smaller after adjustment; we think this is due to the large number of primary study effect sizes that we found were ambiguous.

### Exploratory findings

We extended our preregistration by checking whether irreproducible primary study effect sizes affected the meta-analytic estimate of heterogeneity $\tau^2$. The discrepancies in $\tau^2$ estimates

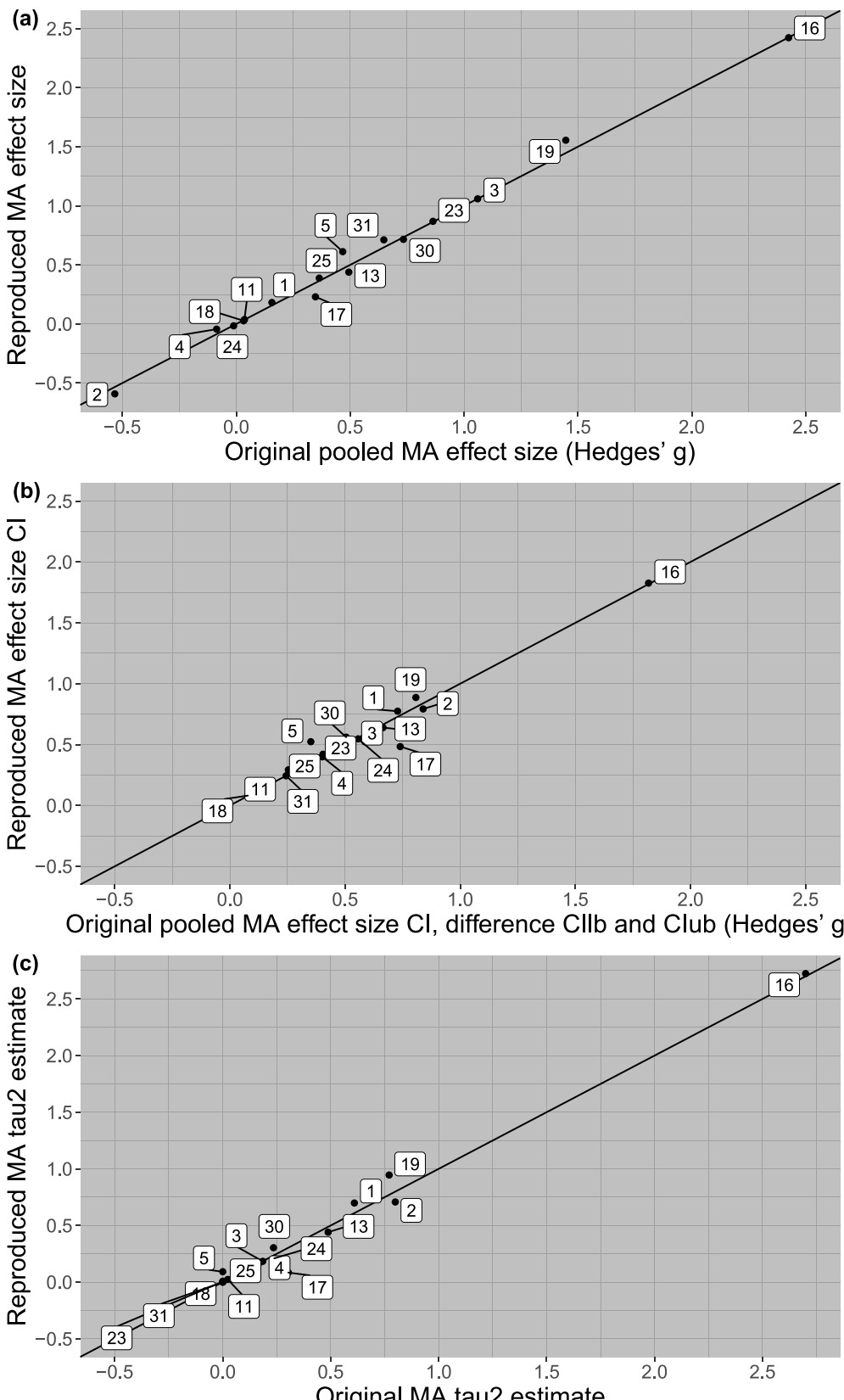

**Fig 5. Scatterplot of reported and reproduced meta-analytic outcomes for meta-analyses using standardized mean differences, where all Cohen's *d* estimates are transformed to Hedges' *g*.**

are displayed in panel *c* of Fig 5 and Fig 6. The heterogeneity estimate changed from statistically significant to non-significant in one meta-analysis, and from statistically non-significant to significant in another meta-analysis. In total, 17 $\tau^2$ parameter estimates were larger after recalculating the primary study effect sizes, 12 were smaller, and four showed no difference.

A reviewer to a previous version of this paper rightfully claimed that it is problematic to include the *incomplete* effect sizes in tests of meta-analytic reproducibility, since these effects could not be reproduced due to a lack of information. Including these incomplete effect sizes might deflate the differences between the reported and reproduced meta-analyses. As such, we decided to compare the reported and reproduced meta-analyses again, only including effect sizes we could calculate: *correct*, *incorrect*, and *ambiguous* effect sizes. In total, 14 out of 33 meta-analyses (42%) showed discrepancies in either the pooled effect size estimate, its confidence interval, or $\tau^2$ parameter, which is one meta-analysis more than in the previous analyses when all four effects were included. The discrepancies between the original and reproduced meta-analyses became larger in this analysis, but the statistical significance of the overall pooled effect sizes did not change; detailed results are displayed in S1 File.

We acknowledge that it is impossible to formulate definite conclusions regarding the reproducibility of full meta-analyses because we took samples of individual effect sizes from each. However, under some strong assumptions we can predict the probability that a full meta-analysis is reproducible. A $\chi^2$ test showed that it is unrealistic to assume that the probability of a reproducible individual effect size is equal across all meta-analyses ($\chi (32) = 156.12$, $p < .001$, $\chi^2$ test of independence). We performed a multilevel logistic regression analysis with (in)correctly reproduced individual effect sizes as the dependent variable, and the 33 meta-analyses as the grouping variable. We used the estimates from this model (intercept = .214, variance = 2.09) to approximate the distribution of the probability of a reproducible effect size with 1,000 points, and used this approximated distribution to calculate the probability of reproducing a meta-analysis with a given number of effect sizes. The analysis script is located at https://osf.io/5k4as/. Using this model, we predict meta-analyses of size 1, 5, 10, 16 or larger to be fully reproducible with respective probabilities .538, .170 .082, and < .050.

## General discussion

In Part 1 we investigated the reproducibility of 500 primary study effect sizes from 33 meta-analyses. In Part 2, we then looked at the effect of irreproducible effect sizes on meta-analytic pooled effect size estimates, confidence intervals and heterogeneity estimates. Almost half of the primary study effect sizes could not be reproduced based on the reported information in the meta-analysis, due to incomplete, incorrect, or ambiguous reporting. However, overall, the consequences were limited for the main meta-analytic outcomes. We found small and a few moderate discrepancies in meta-analytic outcomes in 39% of meta-analyses, but most discrepancies were negligible. In none of the meta-analyses did the use of recalculated primary study effect sizes change the statistical significance of the pooled effect size, whereas the statistical significance of the heterogeneity estimate changed in two meta-analyses. These two meta-analyses were characterized by many ambiguous effect size computations (meta-analysis 5 in Fig 5) and one relatively large effect size discrepancy (meta-analysis 29, Fig 6).

In this study, we focused only on the effect of primary study errors on meta-analytic estimates. Since we found errors in primary studies to have a (minimal) effect on meta-analytic mean and heterogeneity estimates, we expect the errors to also have a (small) effect on

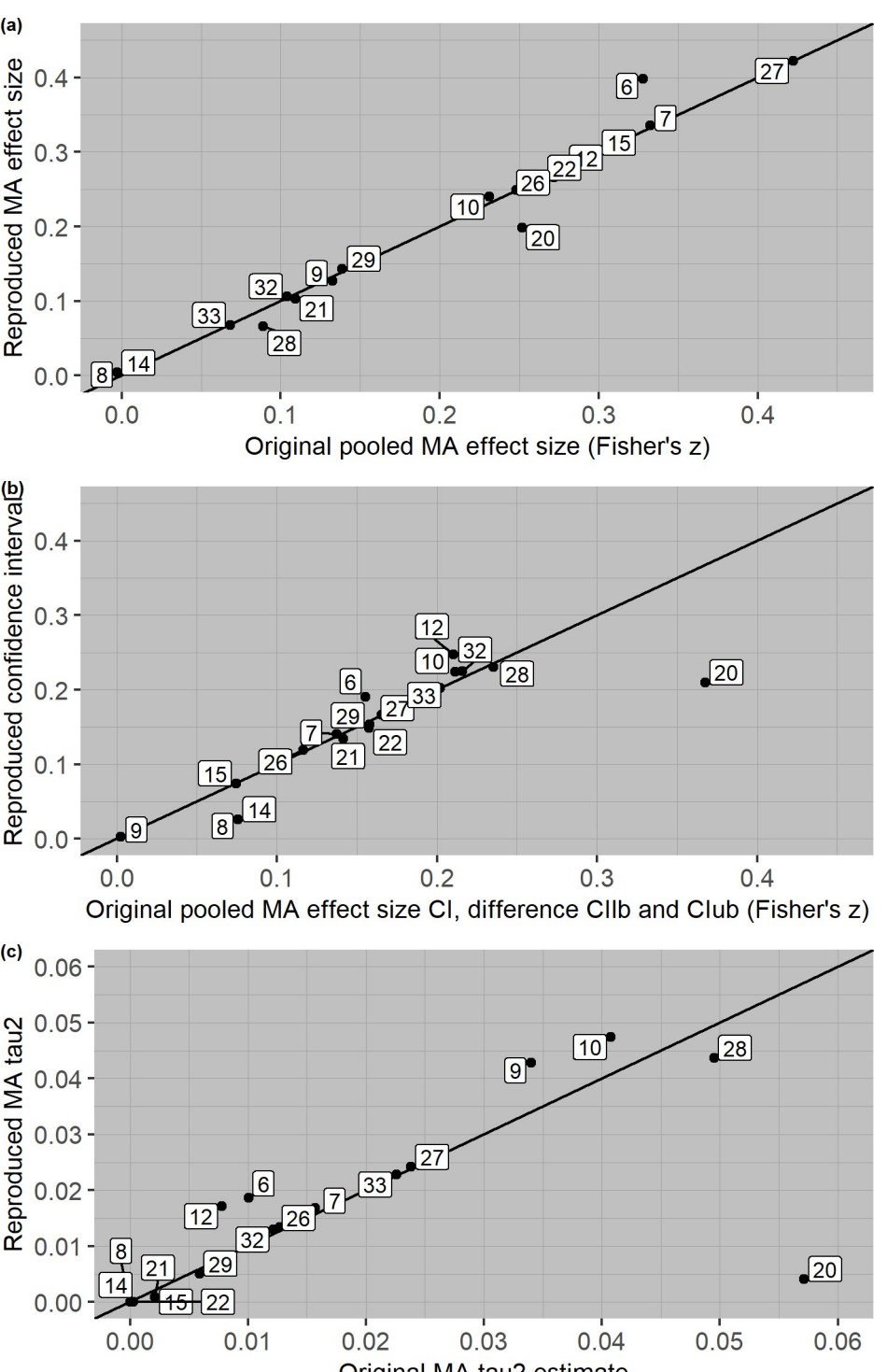

**Fig 6. Scatterplot of reported and reproduced meta-analytic outcomes for meta-analyses using correlations, where all product-moment correlations *r* are transformed to Fisher's *z*.** That the scale of panel c is different from panel a and b.

methods that correct for publication bias. However, as primary study errors seemed mostly random rather than systematic, we expect an increase in variance of estimates when correcting for publication bias. An increase in variation might affect the results by obscuring true patterns of bias, because of diminished power in some analyses of bias. For instance, in Egger's test, the added random variation might lower the power to detect asymmetry in the funnel plot that is indicative of publication bias.

We should note that we were conservative in our estimations for primary study effect sizes for which we did not have enough information from the original papers to recompute them ourselves. We decided to copy those effect sizes as they were reported, meaning they did not count towards any discrepancies when rerunning the meta-analyses in Part 2. Similarly, for ambiguous effects, we chose the estimate that was closest to the reported one, leading to conservative estimates of the discrepancies. We acknowledge that results of our study could have been (very) different if we had decided to include either the minimum, maximum, or a random effect size with these ambiguous effect sizes. Moreover, we took samples from each meta-analysis to keep our coding time manageable, and so we leave it to the interested (and industrious) reader to recompute specific meta-analytic outcomes after checking all effect size computations featured in it. Based on previous research on the reproducibility of meta-analyses in medicine [8], we expect the effect may be significantly more detrimental.

Surprisingly, we found unpublished and non-outlier primary studies to be more reproducible than published or outlier primary studies. It could be that meta-analytic authors are more cautious when calculating effect sizes from unpublished articles because the article is perhaps not peer-reviewed. Similarly, perhaps many meta-analysts do pay more attention to effect sizes that are relatively large.

## Limitations

We recognize some limitations in our study. Our primary study effect size sample is not completely random because we first identified which primary study effect sizes were outliers and oversampled these, before we took a random sample from both outlier and non-outlier primary studies. However, we believe our estimate of the average probability of being able to reproduce a random primary study effect size from a meta-analysis (37%) is accurate, as we corrected for our planned oversampling of outlier primary effect sizes and included a large and systematically drawn sample. Additionally, although our selection of meta-analyses was random and based on a large and fairly comprehensive database, we only included meta-analyses that contained a data table with a minimum amount of information. These meta-analyses can be considered to be relatively well reported compared to meta-analyses lacking any data table. Based on the finding that reluctance to share data is associated with more reporting errors in primary studies ([25]; but see also [26]), one would expect meta-analyses accompanied by open data to be of higher (reporting) quality. Thus, we expect meta-analyses that we omitted because of lacking data to show even weaker reproducibility. If meta-analysts wish to convince readers of reproducible outcomes, they could start sharing their data table and reporting in clear manner how their computations were performed.

Another limitation is that we coded a subset of 500 primary study effect sizes, instead of all 1,978 effects. We acknowledge that it is hard to get a comprehensive view on the reproducibility of meta-analyses based on only a subset of primary study effect sizes. It should however be noted that for 19 out of 33 meta-analyses (57%) we attempted to reproduce at least half of all included primary study effect sizes. Moreover, the coding process for the primary study effect sizes and meta-analyses proved to be very labor intensive, due to the substandard reporting practices that hindered the reproducibility and verifiability of many results. Our study revealed

that most meta-analysts do not adhere to the Meta-Analysis Reporting Standards (MARS; [27]) and Preferred Reporting Items for Systematic reviews and Meta-Analyses (PRISMA) guidelines [16]. This also implies that our reproduced primary study effect sizes might not always have been the primary study effect sizes that were selected by the meta-analytic authors; in many cases (19%) the reporting on which specific effect was extracted from the primary study was vague, and which effects we deemed fitting might differ substantially from what the meta-analysts intended to include. These issues made this project particularly time-consuming, and it is clear that almost none of the published meta-analyses in our sample were easily reproducible, making it nearly impossible to retrace the steps taken during the research.

A final limitation is that our sampled meta-analyses were from 2011 and 2012. It could be that the increasing emphasis on reproducibility in psychology of the last years has incentivized meta-analysts to make reporting more extensive and result checking more diligent. However, results from [15], who checked meta-analyses from 2013–2014, show similar results and emphasize a lack of adherence to reporting standards. More research to adherence of reporting standards and subsequent reproducibility of results in recent meta-analyses is needed to help us understand whether there is any sign of improvement. Meta-analyses are increasingly being used and widely cited. In 2018 alone, the 33 meta-analyses in our sample have been cited a total of 846 times. It is important that future meta-analyses improve in how they report their results.

## Recommendations

The results of this study call for improvement in reporting practices in psychological literature and particularly in meta-analyses. Such improvements entail specifically sharing necessary details regarding the entire meta-analytic process. Meta-analyses should report not only a data table containing the basic primary study statistics, but also details regarding study design and analyses. Specifying conditions, samples, outcomes, variables, and methods used to find and analyze data is imperative for reproducibility. The meta-analytic reporting guidelines such as MARS and PRISMA provide good guidance on meta-analytic reporting [16,27]. Moreover, a recent study found explicit mention of the PRISMA guidelines to be associated with more complete reporting of meta-analyses in psychology [6]. For a meta-analysis to be completely reproducible, we would add to the existing guidelines the requirement that effect size computations should be specified per effect size in supplementary materials. It should be clear which decisions were made and when, and each primary study effect size should be able to be uniquely identified. For an example, we refer to S3 File (https://osf.io/pqt9n), where we documented the relevant text, references, and formulas from all 33 meta-analyses to indicate which specific transformations we made to the effect sizes within meta-analyses. Moreover, in our codebook (https://osf.io/7abwu) we specified the names of the groups and variables that were compared for each primary study effect size, *exactly as they were reported in the primary study*. If certain groups or effects were combined, it was also necessary to add a comment on how and in which order these were combined. We acknowledge it is hard to document all relevant information related to effect size computation in meta-analyses, and emphasize that sharing all data, code, and documentation used in the process would benefit reproducibility of meta-analyses tremendously. For more information on best practices in systematic reviewing, we refer to [28].

Ideally, specification of the methods takes place before the data collection starts to counter biasing effects. By preregistering the meta-analytic methods, an explicit distinction can be made between exploratory and confirmatory analyses. This helps avoid hypothesizing after results are known [29] and potential biases caused by the many seemingly arbitrary choices

meta-analysts make during the collection and processing of data. [30] composed a checklist for various types of researcher degrees of freedom related to psychological research, such as failing to specify the direction of effects and failing to report failed studies. It would be worthwhile to expand this checklist to the context of meta-analyses. Other suggestions for improvement in reporting practices include opening data, materials and workflow to use transparency as an accountability measure (e.g., by dynamic documenting through RMarkdown [31]), or the use of tools that benefit data extraction for systematic reviews [32]. Fortunately, many online initiatives promote preregistration and data sharing practices, with increasingly more journals requiring authors to share the data that would be needed by someone wishing to validate or replicate the research (e.g., PLOS ONE, Scientific Reports, the Open Science Framework, the Dataverse Project).

Accurately conducted and reported meta-analyses are necessary considering the increasing advancement of research and knowledge, and it is crucial that methods in meta-analyses become more reproducible. Only then will trust in meta-analytic results be justified for building better theories, steering future research efforts, and informing practitioners in a wide range of settings.

## Supporting information

**S1 File. Additional results.** https://osf.io/pf4x9/.
(PDF)

**S2 File. Sampling scheme and flowcharts.** https://osf.io/43ju5/.
(PDF)

**S3 File. Formulas and methods.** https://osf.io/pqt9n/.
(PDF)

**S4 File. Reported vs reproduced primary study effect sizes.** https://osf.io/65b8z/.
(PDF)

**S5 File. Post stratification calculations.** https://osf.io/u2j3z/.
(PDF)

## Author Contributions

**Conceptualization:** Esther Maassen, Jelte M. Wicherts.

**Data curation:** Esther Maassen.

**Formal analysis:** Esther Maassen.

**Funding acquisition:** Jelte M. Wicherts.

**Investigation:** Esther Maassen, Anton Olsson-Collentine.

**Methodology:** Esther Maassen, Marcel A. L. M. van Assen, Michèle B. Nuijten, Jelte M. Wicherts.

**Project administration:** Esther Maassen.

**Supervision:** Marcel A. L. M. van Assen, Michèle B. Nuijten, Jelte M. Wicherts.

**Validation:** Esther Maassen, Marcel A. L. M. van Assen, Jelte M. Wicherts.

**Visualization:** Esther Maassen, Jelte M. Wicherts.

**Writing – original draft:** Esther Maassen.

**Writing – review & editing:** Esther Maassen, Marcel A. L. M. van Assen, Michèle B. Nuijten, Anton Olsson-Collentine, Jelte M. Wicherts.

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
