## [Decision Letter · Decision Letter 0]

16 Mar 2020

PONE-D-20-01919

Reproducibility of individual effect sizes in meta-analyses in psychology

PLOS ONE

Dear Ms Maassen,

Thank you for submitting your manuscript to PLOS ONE. After careful consideration, we feel that it has merit but does not fully meet PLOS ONE’s publication criteria as it currently stands. Therefore, we invite you to submit a revised version of the manuscript that addresses the points raised during the review process.

I have now received two reviews from experts in meta-analytic methods. As you can see, both reviewers were very positive and suggested publication of your manuscript. I concur with their assessment. The reviewers raised some minor points that you should clarify in a revision. Let me also add two points of my own:

If I understood it correctly, “reproducible” effect sizes were classified as such if the difference between the original and recalculated effect size was smaller than r = .025 (page 10); so, you were trying to recalculate the effect size with some margin of error. I would recommend clarifying this point when introducing your classification scheme at the end of page 9.It would be informative to describe Figure 4 in more detail. In particular, what was the distribution (e.g., median, quartiles) of the proportion of reproducible effect sizes in the meta-analyses? Is the reported proportion of 37% (page 14) an adequate estimate for all meta-analyses or is the distribution rather skewed; for example, it might be the case that in many meta-analyses most effect sizes are irreproducible, whereas few meta-analyses primarily reported irreproducible effect sizes.

In addition to these points, both reviewers made further excellent suggestions that you should consider in your revision. I strongly encourage you to address these issues and submit a revised version of your manuscript.

We would appreciate receiving your revised manuscript by Apr 30 2020 11:59PM. To enhance the reproducibility of your results, we recommend that if applicable you deposit your laboratory protocols in protocols.io, where a protocol can be assigned its own identifier (DOI) such that it can be cited independently in the future. For instructions see: http://journals.plos.org/plosone/s/submission-guidelines#loc-laboratory-protocols

We look forward to receiving your revised manuscript.

Kind regards,

Timo Gnambs

Academic Editor

PLOS ONE

Journal Requirements:

Reviewers' comments:

Reviewer's Responses to Questions

**Comments to the Author**

1. Is the manuscript technically sound, and do the data support the conclusions?

Reviewer #1: Yes

Reviewer #2: Yes

2. Has the statistical analysis been performed appropriately and rigorously? 

Reviewer #1: Yes

Reviewer #2: Yes

3. Have the authors made all data underlying the findings in their manuscript fully available?

Reviewer #1: Yes

Reviewer #2: Yes

4. Is the manuscript presented in an intelligible fashion and written in standard English?

Reviewer #1: Yes

Reviewer #2: Yes

5. Review Comments to the Author

Reviewer #1: Thank you for the opportunity to review this manuscript. This manuscript is an important contribution to the methodological literature on the conduct and reporting of systematic reviews that include meta-analyses. The manuscript focuses on one of many ways that meta-analyses can fail to be reproducible – the computation of effect sizes. The manuscript makes a clear and compelling case for transparency and the use of open science practices in the conduct and reporting of meta-analysis particularly with regard to the computation of effect sizes. The associated registry of their own conduct of this research project provides an example of how to implement transparency in research.

The manuscript also points out issues that often encountered in reviewing meta-analyses and that reflect issues with current meta-analysis practice. Few reviews adhere to the published standards in MARS or PRISMA. Few published meta-analyses provide complete data tables, much less information on how effect sizes were computed from primary studies. This manuscript should provide more evidence that meta-analyses need to follow current reporting standards and use open science practices.

One hypothesis about why some effect sizes might be irreproducible or ambiguous relates to how the meta-analysis handles multiple effect sizes from primary studies. Since the publication date of the meta-analyses used in this manuscript, methods for handling dependent effect sizes are being used more widely. Prior to the publication of Hedges, Tipton & Johnson (2010), researchers either chose a single effect size from a study or combined effect sizes measuring a similar construct in a non-transparent manner. Though this may be one contributing factor to the irreproducibility of effect sizes in this manuscript, the general message still holds today that more transparency is needed when reporting effect size computations in a meta-analysis. As a minor point, I wondered how many of the irreproducible and ambiguous effect sizes were related to the issue of dependent effect size.

In sum, the manuscript will add to the growing evidence that meta-analyses need to embrace fully open science practices given the dependence of many policy decisions on meta-analytic results.

Reviewer #2: I think that this work is of great interest given the important and lofty role meta-analysis plays in scientists' and practitioners' evaluation of research results. It is useful to finally have some data on the reproducibility of these research results. If I understand the results correctly, the reproducibility of individual effect sizes is poor, but this does not seem to have a substantial or predictable effect on the reproducibility of meta-analytic means and heterogeneity. This is useful to know as we continue to practice meta-analysis, calibrate the degree of trust we put in meta-analysis, and consider which transparency guidelines are most important in meta-analysis.

I can't go through all the data and code, although I appreciate its presence in the Github repository. I loaded up codebook-primary-studies-complete.xlsx just to check on those d > 3 estimates. I guess the SIRS is a very powerful instrument.

This looks like a complete piece of work that has already been through several rounds of peer review. I only have some minor suggestions:

It's not clear to me that "ambiguous" is necessarily worse than "incorrect". As introduced in line 190 I thought this was going to be the basis for some sort of ordinal regression but was relieved to see it is merely descriptive and the badness of these categories merely a statement of the authors' beliefs.

Discussion

When you say "For a meta-analysis to be completely reproducible, we would add to these guidelines the requirement that effect size computations should be specified per effect size in supplementary materials. It should be clear which decisions were made and when, and each primary study effect size should be able to be uniquely identified." Can you point to examples in your own codebook to provide a concrete demonstration of how this should look?

Future directions:

Authors will sometimes confuse SE and SD in their own writing, leading to meta-analysts using the SE instead of the SD when calculating effect sizes, yielding effects that are too big. Did you observe this with any frequency in this data set? This could yield an effect size that is reproducible given the primary study and yet incorrect.

These errors seem to have little influence over the mean and heterogeneity. How might these errors influence less robust statistics like PET/PEESE, p-curve, etc?

I always sign my reviews,

Joe Hilgard

6. PLOS authors have the option to publish the peer review history of their article (what does this mean?). If published, this will include your full peer review and any attached files.

Reviewer #1: Yes: Terri Pigott

Reviewer #2: Yes: Joseph Hilgard

---

## [Author Response · Author response to Decision Letter 0]

26 Apr 2020

April 26, 2020

Dear Dr. Gnambs,

We hereby resubmit our manuscript ‘Reproducibility of Individual Effect Sizes in Meta-Analyses in Psychology’. We have responded to all points raised by you and the reviewers. Below, we respond to all the points in italics, and we have added the specific edits we made to the manuscript after in red.

We hope our revision has addressed the comments of both you and the reviewers sufficiently. We look forward to your reply.

Kind regards, 

Esther Maassen 

also on behalf of, Marcel A.L.M. van Assen, Michèle B. Nuijten, Anton Olsson-Collentine, and Jelte M. Wicherts. 

emaassen@protonmail.com

------ 

1. If I understood it correctly, “reproducible” effect sizes were classified as such if the difference between the original and recalculated effect size was smaller than r = .025 (page 10); so, you were trying to recalculate the effect size with some margin of error. I would recommend clarifying this point when introducing your classification scheme at the end of page 9.

As per your recommendation, we have clarified there is a margin of error when calculating {ir}reproducible effect sizes. We have now explicitly stated this margin for the two most common effect sizes (correlation r and standardized mean difference Hedges' g) on page 9 and 10: 

We categorized reproduced primary study effects into one of four categories we considered, ranging from best to worst outcome: (0) reproducible: we could reproduce the effect size as reported in the meta-analytic article or with a margin of error of correlation r < .025 or Hedges’ g < .049; (1) incomplete: not enough information was available to reproduce the effect size (e.g., SDs are missing in the primary article). In this case we copied the original effect size as reported in the meta-analysis because we were not sure what computations the meta-analysts performed, or whether they had contacted the authors for necessary statistics; (2) incorrect: our recalculation resulted in a different effect size of at least r ≥ .025 or Hedges’ g ≥ .049 (i.e., a potential calculation or reporting error was made);

2. It would be informative to describe Figure 4 in more detail. In particular, what was the distribution (e.g., median, quartiles) of the proportion of reproducible effect sizes in the meta-analyses? Is the reported proportion of 37% (page 14) an adequate estimate for all meta-analyses or is the distribution rather skewed; for example, it might be the case that in many meta-analyses most effect sizes are irreproducible, whereas few meta-analyses primarily reported irreproducible effect sizes.

We have described the distribution of effect size errors in more detail on page 15. We added percentages per category per meta-analysis to Figure 4. We added the following:

Fig 4 displays a bar plot with the frequency of irreproducible effect sizes per meta-analysis. The distribution of reproducible effect sizes (category 0) ranged from 0% to 100% with a mean of 53% and median of 56%. Only three of the 33 samples of primary studies were completely reproducible, and one was completely irreproducible (k = 11). The percentage of incorrect effect sizes (category 1) ranged from 0% to 91% across meta-analyses, (mean = 14%, median = 11%); incomplete effect sizes ranged from 0% to 67% (mean = 12%, median = 5%), and ambiguous effect sizes (category 3) ranged from 0% to 91% (mean = 19%, median = 11%). Note that the reporting within meta-analyses is often at least partly ambiguous (24 out of 33 meta-analyses contain at least one ambiguous effect size).

3. As a minor point, I wondered how many of the irreproducible and ambiguous effect sizes were related to the issue of dependent effect size.

Reviewer 1 recommended to add more details on the reproducibility rates when using dependent effect sizes (i.e., when multiple effects within one primary study are combined). We did so on page 13, and added a new table (Table 1) containing percentages of effect size reproducibility for single and multiple effect sizes. More specifically, we added the following paragraph:

Within over a quarter of primary studies (147; 29%, see Table 1) we combined multiple effects into one overall effect size estimate for that primary study. The percentage of irreproducible effect sizes is relatively large within this group. Within this subset, 18% was classified as incorrect (single effect sizes: 13%), 34% as ambiguous (single effect sizes: 13%), 1% as incomplete (single effect sizes: 15%), and 46% as reproducible (single effect sizes: 59%); Χ^2(3, N = 500) = 45.78, p < .0001, ΦCramer = 0.30, showing that combining multiple effect sizes into one overall estimate is moderately associated with irreproducibility of effect size estimates. 

Table 1. Reproducibility frequencies separated by primary study effect sizes consisting of one or multiple combined effect sizes

Single effect size Combined effect sizes

Reproducible 208 (59%) 68 (46%)

Incorrect 47 (13%) 27 (18%)

Incomplete 52 (15%) 2 (1%)

Ambiguous 46 (13%) 50 (34%)

Total 353 (100%) 147 (100%)

4. It's not clear to me that "ambiguous" is necessarily worse than "incorrect". As introduced in line 190 I thought this was going to be the basis for some sort of ordinal regression but was relieved to see it is merely descriptive and the badness of these categories merely a statement of the authors' beliefs.

On page 10 we clarified that the categorization of effect sizes is our own, and is only used descriptively. More specifically:

We consider ambiguous effects (category 3) to be more problematic than incorrect effects (category 2), as for the incorrect effects it was clear which effect size the meta-analysts had chosen, whereas for the ambiguous effects it was impossible to ascertain if we had selected the correct effect size, and what steps were taken to get to the effect size as reported. We acknowledge this categorization is open to discussion, and only used it descriptively throughout this study.

5. When you say "For a meta-analysis to be completely reproducible, we would add to these guidelines the requirement that effect size computations should be specified per effect size in supplementary materials. It should be clear which decisions were made and when, and each primary study effect size should be able to be uniquely identified." Can you point to examples in your own codebook to provide a concrete demonstration of how this should look?

Reviewer 2 asked us to point to an example in our own codebook to provide a concrete demonstration on how specific effect size computation should look like. We added an extra reference to our supplemental materials in text, which indicate all inclusion criteria and formulas used per meta-analysis, as well as our codebook that documents the relevant information per primary study. More specifically: 

For a meta-analysis to be completely reproducible, we would add to the existing guidelines the requirement that effect size computations should be specified per effect size in supplementary materials. It should be clear which decisions were made and when, and each primary study effect size should be able to be uniquely identified. For an example, we refer to S3 (https://osf.io/pqt9n/), where we documented the relevant text, references, and formulas from all 33 meta-analyses to indicate which specific transformations we made to the effect sizes within meta-analyses. Moreover, in our codebook (https://osf.io/7abwu/) we specified the names of the groups and variables that were compared for each primary study effect size, exactly as they were reported in the primary study. If certain groups or effects were combined, it was also necessary to add a comment on how and in which order these were combined. We acknowledge it is hard to document all relevant information related to effect size computation in meta-analyses, and emphasize that sharing all data, code, and documentation used in the process would benefit reproducibility of meta-analyses tremendously. For more information on best practices in systematic reviewing, we refer to [28].

6. Authors will sometimes confuse SE and SD in their own writing, leading to meta-analysts using the SE instead of the SD when calculating effect sizes, yielding effects that are too big. Did you observe this with any frequency in this data set? This could yield an effect size that is reproducible given the primary study and yet incorrect.

Reviewer 2 stated that a common mistake made by authors is confusing the SE and SD. We indicated on page 12 and 13 how often this occurred in our set of primary studies, and added estimates for other occurrences (e.g., how often it was unclear which variables to include in the primary study effect size estimation). More specifically:

The most common reason for not being able to reproduce a primary study effect size was missing or unclear information in the meta-analysis (i.e., ambiguous effect sizes, k = 96, 19%). More specifically, it was often unclear which specific effect was extracted from the primary study because multiple effects were relevant to the research question, and we did not know if and how the included effect was constructed from a combination of multiple effects. Other prevalent issues pertaining to potential data errors or lack of clarity in the meta-analytic process were inconsistencies and unclear reporting in inclusion criteria within a meta-analytic article (k = 67), uncertainty about which samples or time points were included (k = 50), reporting formulas or corrections that were incorrect or not used (k = 23), including the same sample of respondents for multiple effects without correction (k = 7), lacking information on how the primary study effect was transformed to the effect size included in the meta-analysis (k =3), and mistaking the standard error for the standard deviation when calculating effects (k = 2). 

7. These errors seem to have little influence over the mean and heterogeneity. How might these errors influence less robust statistics like PET/PEESE, p-curve, etc?

Reviewer 2 asked how the reproducibility errors might influence less robust statistics like PET/PEESE and p-curve. Since our focus in this study was based on meta-analytic outcomes, we decided not to check for each of the 33 meta-analyses separately what the effects of these errors on publication bias methods were. We added a paragraph to our discussion section on page 22 on what we expect the effect on publication bias methods to be. More specifically:

In this study, we focused only on the effect of primary study errors on meta-analytic estimates. Since we found errors in primary studies to have a (minimal) effect on meta-analytic mean and heterogeneity estimates, we expect the errors to also have a (small) effect on methods that correct for publication bias. However, as primary study errors seemed mostly random rather than systematic, we expect an increase in variance of estimates when correcting for publication bias. An increase in variation might affect the results by obscuring true patterns of bias, because of diminished power in some analyses of bias. For instance, in Egger’s test, the added random variation might lower the power to detect asymmetry in the funnel plot that is indicative of publication bias.

---

## [Editor Report · Decision Letter 1]

29 Apr 2020

Reproducibility of individual effect sizes in meta-analyses in psychology

PONE-D-20-01919R1

Dear Dr. Maassen,

We are pleased to inform you that your manuscript has been judged scientifically suitable for publication and will be formally accepted for publication once it complies with all outstanding technical requirements.

With kind regards,

Timo Gnambs

Academic Editor

PLOS ONE
---

## [Editor Report · Acceptance letter]

4 May 2020

PONE-D-20-01919R1 

Reproducibility of individual effect sizes in meta-analyses in psychology 

Dear Dr. Maassen:

I am pleased to inform you that your manuscript has been deemed suitable for publication in PLOS ONE. Congratulations! Your manuscript is now with our production department. 

With kind regards,

on behalf of

Dr. Timo Gnambs 

Academic Editor

PLOS ONE